# Measures of genetic diversification in somatic tissues at bulk and single-cell resolution

**Marius E Moeller[1†], Nathaniel V Mon Père[2,3†], Benjamin Werner[2]\*, Weini Huang[1,4]\***

[1]Department of Mathematics, Queen Mary University of London, London, United Kingdom; [2]Evolutionary Dynamics Group, Centre for Cancer Genomics and Computational Biology, Barts Cancer Centre, Queen Mary University of London, London, United Kingdom; [3]Interuniversity Institute of Bioinformatics in Brussels, Université Libre de Bruxelles, Ixelles, Belgium; [4]Group of Theoretical Biology, The State Key Laboratory of Biocontrol, School of Life Science, Sun Yat-sen University, Guangzhou, China

**\*For correspondence:**
b.werner@qmul.ac.uk (BW);
weini.huang@qmul.ac.uk (WH)

†These authors contributed equally to this work

**Competing interest:** The authors declare that no competing interests exist.

**Abstract** Intra-tissue genetic heterogeneity is universal to both healthy and cancerous tissues. It emerges from the stochastic accumulation of somatic mutations throughout development and homeostasis. By combining population genetics theory and genomic information, genetic heterogeneity can be exploited to infer tissue organization and dynamics in vivo. However, many basic quantities, for example the dynamics of tissue-specific stem cells remain difficult to quantify precisely. Here, we show that single-cell and bulk sequencing data inform on different aspects of the underlying stochastic processes. Bulk-derived variant allele frequency spectra (VAF) show transitions from growing to constant stem cell populations with age in samples of healthy esophagus epithelium. Single-cell mutational burden distributions allow a sample size independent measure of mutation and proliferation rates. Mutation rates in adult hematopietic stem cells are higher compared to inferences during development, suggesting additional proliferation-independent effects. Furthermore, single-cell derived VAF spectra contain information on the number of tissue-specific stem cells. In hematopiesis, we find approximately $2 \times 10^5$ HSCs, if all stem cells divide symmetrically. However, the single-cell mutational burden distribution is over-dispersed compared to a model of Poisson distributed random mutations. A time-associated model of mutation accumulation with a constant rate alone cannot generate such a pattern. At least one additional source of stochasticity would be needed. Possible candidates for these processes may be occasional bursts of stem cell divisions, potentially in response to injury, or non-constant mutation rates either through environmental exposures or cell-intrinsic variation.

## eLife assessment

In this paper, the authors introduce **fundamental** work on mathematical methods for inferring evolutionary parameters of interest from RNA data in healthy tissue and during hematopoiesis. By combining single cell and bulk sequencing analyses, the authors use a stochastic process to inform different aspects of genetic heterogeneity; the strength of evidence in support of the authors' claim is **exceptional**. The work will be of broad interest to cell biologists and theoretical biologists.

## Introduction

Intra-tissue genetic heterogeneity emerges from a multitude of dynamical processes (*Turajlic et al., 2019*; *Black and McGranahan, 2021*). Cells randomly accumulate mutations, they self-renew, differentiate, or die and clones of varying fitness may compete and expand (*Watson et al., 2020*; *Rulands*

*et al., 2018*; *Werner et al., 2020*; *Gunnarsson et al., 2021*). These core evolutionary principles form the basis for describing the aging of somatic tissues, tumor initiation, and tumor progression (*Martincorena, 2019*; *Greaves and Maley, 2012*; *Cagan et al., 2022*; *Martincorena et al., 2018*; *Mitchell et al., 2022*; *Abascal et al., 2021*). Naturally, there is great interest in understanding precisely how these dynamical processes act. This typically involves hypothesizing models of cell behavior, and deriving quantitative estimates of their underlying physical parameters (*Watson et al., 2020*; *Werner et al., 2020*; *Poon et al., 2021*; *Martincorena et al., 2017*; *Chatzeli and Simons, 2020*; *Williams et al., 2020*; *Durrett, 2013*). These could by quantities such as mutation rates per cell division, active stem cell numbers, symmetric and asymmetric division rates, etc.,. Some such parameters are already well characterized in certain tissues. We know for example that during early development healthy somatic cells accumulate on average between 1.2 and 1.3 mutations per genome per division (*Werner et al., 2020*; *Lee-Six et al., 2018*). Other parameters, such as the number of stem cells in a tissue type, are well known in some tissues but harder to quantify in others.

In general, a lack of time-resolved information complicates evolutionary inferences, and we must often rely on indirect information, for example, the patterns of tissue-specific genetic heterogeneity within and across individuals (*Bailey et al., 2021*). These observed patterns of genetic heterogeneity emerge from the underlying stochastic processes and can strongly depend on the measurement technique (*Caravagna et al., 2020*). Both enforce specific limitations on the inferability of the underlying dynamics. Different types of genomic data – such as bulk and single-cell whole genome sequencing – may contain different information on the system, and each comes with their own limitations. When bulk sequencing data is used for inference, some evolutionary parameters such as the population size and proliferation rates of stem cells are entangled and cannot be obtained in isolation (*Watson et al., 2020*; *Williams et al., 2020*). If on the other hand, single-cell data is employed, some additional information becomes available, for example the individual cell mutational burdens and the co-occurrence of mutations. However, often the sampling size is sparse compared to the presumed number of long-term proliferating cells in many tissues (*Lee-Six et al., 2018*; *Salehi et al., 2021*; *Lim et al., 2020*), which in turn introduces other constraints. Here, we show that combining such data obtained from different resolutions (bulk or single-cell) can help to overcome limitations and further narrow possible ranges of inferred parameters. Here, we develop concrete mathematical and computational models for extracting evolutionary parameters from bulk and single-cell information and apply these methods to whole genome bulk sequencing data in healthy esophagus (*Martincorena et al., 2018*) and whole genome single-cell sequencing data in healthy hematopoiesis (*Lee-Six et al., 2018*).

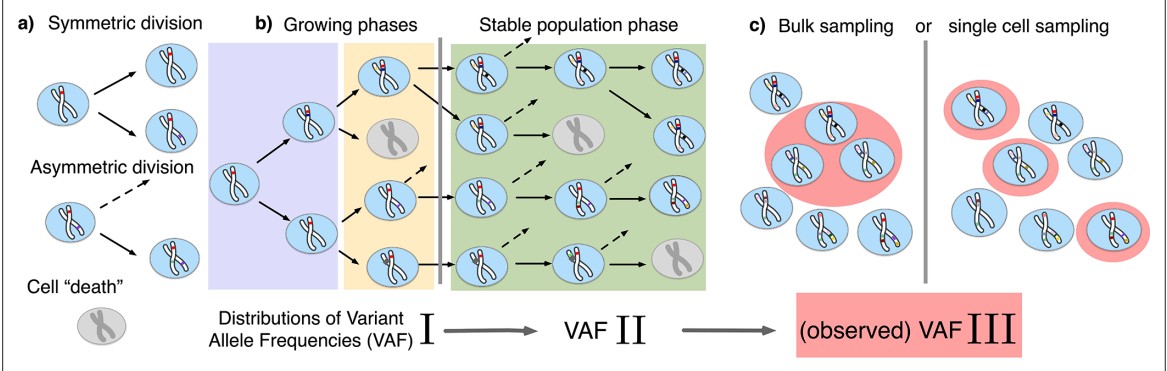

**Figure 1.** The distribution of variant allele frequencies changes with the growth phases and by sampling. (**a**) In the current population, cells divide symmetrically into two daughter cells or asymmetrically with only one daughter cell kept in the focused population. All other events are mathematically equivalent to and are treated as a part of cell death. (**b**) The rates of symmetric and asymmetric division change during the population growth and lead to a dynamic distribution of variant allele frequencies. (**c**) The observed VAF distribution is shifted again during sampling compared to the VAF of the whole population – a fact should be considered when inferring population properties through genetic data.

# Results

## A stochastic model of mutation accumulation in healthy somatic tissue

We model the stem cell dynamics of healthy tissue as a collection of individual cells that divide, differentiate, and die stochastically at predefined rates (see *Figure 1*). Novel mutations can occur with every cell division, each daughter acquiring a random number drawn from a Poisson distribution with rate $\mu$. We explicitly include symmetric and asymmetric stem cell divisions at potentially different rates, the former resulting in two stem or two differentiated cells, and the latter in one stem and one differentiated cell. Since differentiated cells are lost from the stem cell pool they are functionally equivalent to cell death. While only symmetric divisions allow variants to change in frequency, both division types introduce novel variants into the population and thus both contribute to the evolving heterogeneity.

Mutation accumulation occurs through all developmental stages from conception to death (*Spencer Chapman et al., 2021*). We, therefore, consider three phases of demographic changes of the stem cell population. In the first early developmental phase the stem cell population rapidly expands from a single-cell exclusively by symmetric divisions at rate $\gamma$. This is followed by a growth and maintenance phase wherein the population continues to expand but also undergoes turnover through asymmetric divisions at rate $\phi$, as well as cell removal (due to differentiation or death) and replacement at rate $\rho$. In the final mature phase, cell turnover continues but the size of the total stem population $N$ remains constant.

We employ two independent implementations of our theoretical model. One is a direct stochastic simulation utilizing a Gillespie algorithm. This approach creates a simulated dataset containing all single-cell or bulk sequencing properties, however, it can be computationally expensive. In addition, we compute the time-dynamical expected value of the distribution of variant allele frequencies (the VAF spectrum) directly, which we find obeys the partial differential equation

$$\partial_t V(\kappa, t) = -\partial_\kappa \mathcal{A}(\kappa, t) V(\kappa, t) + \partial_\kappa^2 \mathcal{B}(\kappa, t) V(\kappa, t) + \mathcal{C}(t)\delta(\kappa - 1), \tag{1}$$

where $\kappa = fN(t)$ denotes the number of cells sharing a variant (the variant frequency $f$ times the total population size $N$), $\delta(x)$ is the Dirac impulse function, $\partial_t$, and $\partial_\kappa$ are the partial derivatives with respect to time and variant size, and

$$\mathcal{A}(\kappa, t) = \gamma\kappa, \qquad \mathcal{B}(\kappa, t) = \kappa(1 - \kappa/N(t))\rho + \gamma\kappa/2, \qquad \mathcal{C}(t) = 2\mu N(t)(\rho + \gamma + \phi/2),$$

This set of equations allows for computationally efficient numerical solutions (SI 1.3).

## Transition of developmental growth and constant population size signatures in bulk whole genome sequencing data of healthy adult esophagus

We first discuss established properties of the VAF spectrum relating to somatic tissues. In certain model systems it is given by a power law $\propto f^{-\alpha}$ with a critical exponent $\alpha$. The value of the exponent depends on the demographic dynamics of the population. For a well-mixed exponentially growing population without cell death the VAF spectrum $v(f)$ is given by $2\mu/(f + f^2)$ (a $f^{-2}$ power law) and is independent of time (*Gunnarsson et al., 2021*). In contrast, for a population of constant size – i.e., where birth and death rates are equal – the spectrum obeys $v(f) \propto 2\mu/f$ (*Durrett, 2015*) (a $f^{-1}$ power law; see also SI 1.5.1), though this solution is only valid at sufficiently long times. In the following, we focus on the dynamics of genetic diversification in healthy tissues and show that the expected VAF spectrum contains both contributions of growing and constant population dynamics.

Healthy adult somatic tissues are thought to initially expand rapidly during ontogenic growth, continue to expand during infancy and childhood, and reach homeostasis by adulthood. It is thus natural to ask what the expected VAF spectrum would look like in tissues experiencing homeostasis after a period of growth. We first investigated the underlying VAF dynamics in a minimal theoretical model where the population switches from pure exponential growth to constant size once a maximal cell number is reached. Numerical solutions of *Equation 1* show that the expected VAF distribution exhibits a gradual transition from the $f^{-2}$ (growing population) to the $f^{-1}$ (constant population) power law (*Figure 2a*). These transitional states themselves do not adhere to some intermediate power law (e.g. $f^{-\alpha}$ for $1 < \alpha < 2$), but instead present a sigmoidal shape, with the low-frequency portion following $f^{-1}$ and the high frequencies $f^{-2}$. Over time the shape changes as a wavelike front

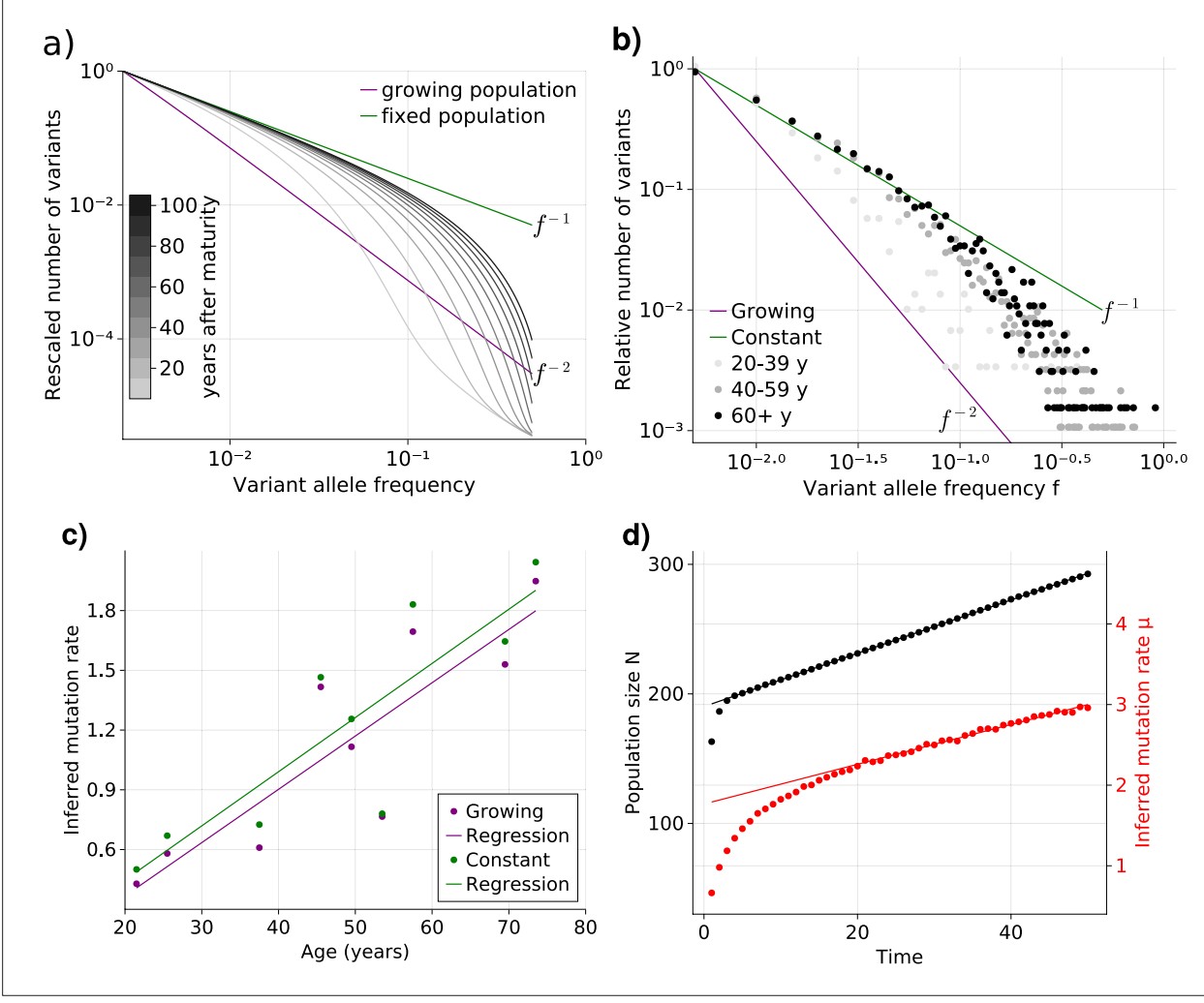

**Figure 2.** Bulk sequencing based variant allele frequency (VAF) and mutation rate inferences in healthy esophagus. (**a**) Expected VAF distributions from evolving *Equation 1* to different time points for a population with an initial exponential growth phase and subsequent constant population phase (mature size $N = 10^3$). Once the population reaches the maximum carrying capacity, the distribution moves from a $1/f^2$ growing population shape (purple) to a $1/f$ constant population shape (green). Note that the shift slows considerably at older age. (**b**) VAF from healthy tissue in the esophagus of nine individuals sorted into age brackets. The youngest bracket, 20–39, is closer to the developmental $1/f^2$ scaling. The older age brackets are both close to the constant population $1/f$ scaling, resembling the theoretical expectations. (**c**) Inferred mutation rates increase linearly with age. (**d**) Simulations of slowly growing stem cell populations reveal that mutation rates appear to increase with age, although the true underlying per division mutation rate remaining constant (see *Figure 2—figure supplement 1* as well).

The online version of this article includes the following figure supplement(s) for figure 2:

**Figure supplement 1.** Comparison of an exponential and logistic growth model.

traveling from low to high frequency, with the constant-size equilibrium establishing earliest at the lowest frequencies and moving to higher frequency overtime. Interestingly, the convergence towards equilibrium slows down over time – for evenly-spaced observation times the solutions lie increasingly closer together – further decreasing the speed at which the high frequency portion of the spectrum approaches equilibrium. More complex forms of such demographic changes, for example, the inclusion of a mixed growth-and-maintenance phase or a logistically growing population, result in qualitatively similar behavior (see *Figure 2—figure supplement 1*).

To test whether adult human tissues show such transitional signatures, we grouped the VAFs of healthy adult esophagus samples from Martincorena and colleagues (*Martincorena et al., 2018*) into three age groups (young, middle. and old, *Figure 2b*). In accordance with our theoretical expectations,

the averaged VAF spectrum of the young donor group is closest to the expected $f^{-2}$ distribution of ontogenic growth, with low frequency variants only starting to approach the $f^{-1}$ homeostatic scaling. The averaged spectra of the middle and old age groups on the other hand notably shift towards the expected $f^{-1}$ homeostatic line. Interestingly, while there is a clear separation between the spectra of the youngest and middle age groups, those of the middle and older groups are much closer. This agrees with our prediction that the speed of convergence of the spectra towards homeostatic scaling slows down with age.

## VAF-based mutation rate inferences in healthy tissues

The theoretical prediction for the VAF spectrum of an exponentially growing population, which is recovered in many cancers given sufficient sequencing depth (*Caravagna et al., 2020*), contains information on the effective mutation rate $\mu$ (*Williams et al., 2016*). Similarly, the VAF spectrum of a constant population includes the same mutation rate $\mu$ (SI). Thus we can in principle infer $\mu$ from the previously described esophagus data, for example by applying a regression approach. However, since the scaling of the VAF spectrum of healthy tissues can be shown to be age-dependent (*Figure 2a–b*), we use two reference shapes for a growing and a constant population, respectively. The fitting is performed by a linear regression least-squares approach. The general solution is computed from the VAF spectrum of the data, $V_d(\mu)$, and the reference shape $V_r$, which assumes $\mu = 1$, as follows:

$$\mu = \frac{\mathrm{Cov}(V_d(\mu), V_r)}{\mathrm{Var}(V_r)} \tag{2}$$

If $V_d(\mu)$ has the shape of $V_r$, then $\mathrm{E}(V_d(\mu)) = \mu\,\mathrm{E}(V_r)$, which can be used to prove *Equation 2*:

$$\mathrm{Cov}(V_d(\mu), V_r) = \mathrm{E}(V_d(\mu), V_r) - \mathrm{E}(V_d(\mu))\,\mathrm{E}(V_r) = \mu(\mathrm{E}(V_r^2) - \mathrm{E}(V_r)^2) = \mu \cdot \mathrm{Var}(V_r) \tag{3}$$

The VAF shapes for the growing and constant population give close estimates, as can be seen in *Figure 2*, which compares the age of all individuals with corresponding inferred effective mutation rates. The estimates are in the range of 1–2 mutations per genome per cell division and agree with the previous observations based on early developmental stem cell divisions (*Lee-Six et al., 2018*). Estimates based on a growing population are slightly higher compared to estimates based on a constant population, but differences are small.

Surprisingly, we observe a clear trend of a linearly increasing effective mutation rate with age. This trend cannot be explained by the transition of variants from a growing into a constant population alone. An actual increase in the true mutation rate per cell division with age would be a direct explanation. However, this seems unlikely: It is well established that the mutational burden across individuals and tissues increases linearly with age (*Cagan et al., 2022*; *Mitchell et al., 2022*; *Williams et al., 2022*), and an increasing mutation rate would in contrast result in an accelerated increase of mutational burden with age, which is not observed experimentally.

An alternative explanation is a continued slow linear increase of the stem cell population with age (*Figure 2d*). In such a scenario, the effective mutation rate would appear to increase linearly with age as well, although the true mutation rate per cell division remains unchanged. At first glance, a linearly increasing stem cell population appears unnatural. However, such a linear increase would emerge from a small bias of stem cells towards self-renewal (growth) and concomitantly a slowdown of the stem cell proliferation rate proportional to the change in population size (*Werner et al., 2015*). The latter is a natural feedback mechanism ensuring a constant output of differentiated somatic cells. Direct in vivo imaging in multiple human tissues including esophagus seem to support an increased density and slower proliferation of stem cells with age (*Tomasetti et al., 2019*). This explanation is attractive for another reason: A feedback of decreased cell proliferation with increasing population size would also function as a tumor suppressor mechanism, and maybe another reason cancer driver mutations in healthy tissues are abundant but rarely progress to cancer (*Martincorena et al., 2018*;

*Martincorena et al., 2015*). Progression would require additional genomic or environmental changes to overcome this regulatory feedback.

## Single-cell mutational burden allows sample size independent inferences of stem cell proliferation and mutation rates

With the increasing availability of single-cell data, it becomes possible to investigate distributions of certain quantities for which bulk sequencing only provides average measurements. One such is the number of mutations present in individual cells. Recent work by Lee-Six and colleagues (*Lee-Six et al., 2018*) showed that in a 59-year-old donor hematopoietic stem cells had accumulated on average around 1000 mutations per cell; however, there was significant variation between individual genomes, some carrying 900 mutations, while others up to 1200. This variation is the result of the stochastic nature of mutation accumulation and cell division, and can be exploited to estimate per division mutation and stem cell proliferation rates. To this end, we look at the distribution of the number of mutations per cell, which we will from here on refer to as the *mutational burden distribution*. Formally, we describe the mutational burden $m_j$ in a cell $j$ by a stochastic sum over its past divisions $m_j = \sum_i^{y_j} u_{ij}$, in which both the total number of divisions $y_j$ in the cell's past and the number of mutations per division $u_{ij}$ are random variables. Even without knowing the actual distributions of $y$ and $u$, general expressions for the expectation and variance of $m_j$ that only depend on the expectation and variance of $y$ and $u$ exist and are given by $\mathrm{E}(m) = \mathrm{E}(y)\,\mathrm{E}(u)$ and $\mathrm{Var}(m) = \mathrm{E}(y)\,\mathrm{Var}(u) + \mathrm{E}(u)^2\,\mathrm{Var}(y)$ (SI 1.7). These expressions allow direct estimates of the effective division and mutation rate. Assuming the mutation rate per daughter cell is Poisson distributed with expectation $\mu$, the average number of divisions per lineage — expressed in terms of an effective division rate $\lambda$ — becomes

$$\int \lambda(t)dt = \mathrm{E}(y) = \mathrm{E}(m)/\mu \tag{4}$$

In addition, if the mutation rate $\mu$ is unknown, it can similarly be obtained from

$$\mu = \left( \frac{\mathrm{Var}(m)}{\mathrm{E}(m)} - 1 \right) \frac{\mathrm{E}(y)}{\mathrm{Var}(y)}. \tag{5}$$

We first show that these estimators recover per division mutation and proliferation rates in stochastic simulations (*Figure 3a and b*). More importantly, they only require information on comparably few single cells. Approximately, 100 cells are sufficient to reliably reconstruct the mutational burden distribution, which for example would constitute a small sample of the hematopoietic stem cell pool. In fact, the inference does not significantly improve if we increase data resolution beyond a few hundred single cells (*Figure 3a and b*). This is in stark contrast to inferences based on single-cell phylogeny: Although they contain more information in principle, important aspects of a single-cell phylogeny, e.g., the number of observable branchings are strongly affected by sampling and stem cell population size.

We apply our estimators to single-cell mutational data in blood obtained from the study by *Lee-Six et al., 2018*, in which 89 hematopoietic stem cells (HSCs) were extracted from a bone marrow aspirate of a 59-year-old individual (*Figure 4a*). Using *Equation 4* to estimate the expected number of divisions per lineage from the sample burden (SI 1.8.1), we find a total division rate (including both symmetric and asymmetric divisions) in the constant population phase of $\lambda = 10.6$ (7.6-15.3) divisions per HSC per year, which is within ranges previously suggested (*Watson et al., 2020*; *Dingli and Pacheco, 2006*). Notably, the single-cell mutational burden alone cannot disentangle symmetric and asymmetric division rates. Applying *Equation 5* to the sample burden distribution and taking $\mathrm{Var}(y)/\mathrm{E}(y) = 1$ (akin to assuming exponentially distributed division times, a common but simplified null model) we obtain an estimated mutation rate of $\tilde{\mu} = 4.3$ per cell per division (*Figure 4a*). This is significantly higher compared to the 1.2 mutations per division suggested previously (*Werner et al., 2020*; *Lee-Six et al., 2018*). We posit different possible explanations for this discrepancy. Stem cell divisions and mutation accumulation are stochastic processes, therefore variation between individuals is expected. However, simulations of our model for a wide range of parameter values suggest the distribution of these inferred mutation rates is unlikely to present a wide enough variance to explain this difference (*Figure 3—figure supplement 2*). Recently, Mitchell and colleagues (*Abascal et al., 2021*) suggested that many mutations in somatic tissue are acquired independently of cell divisions.

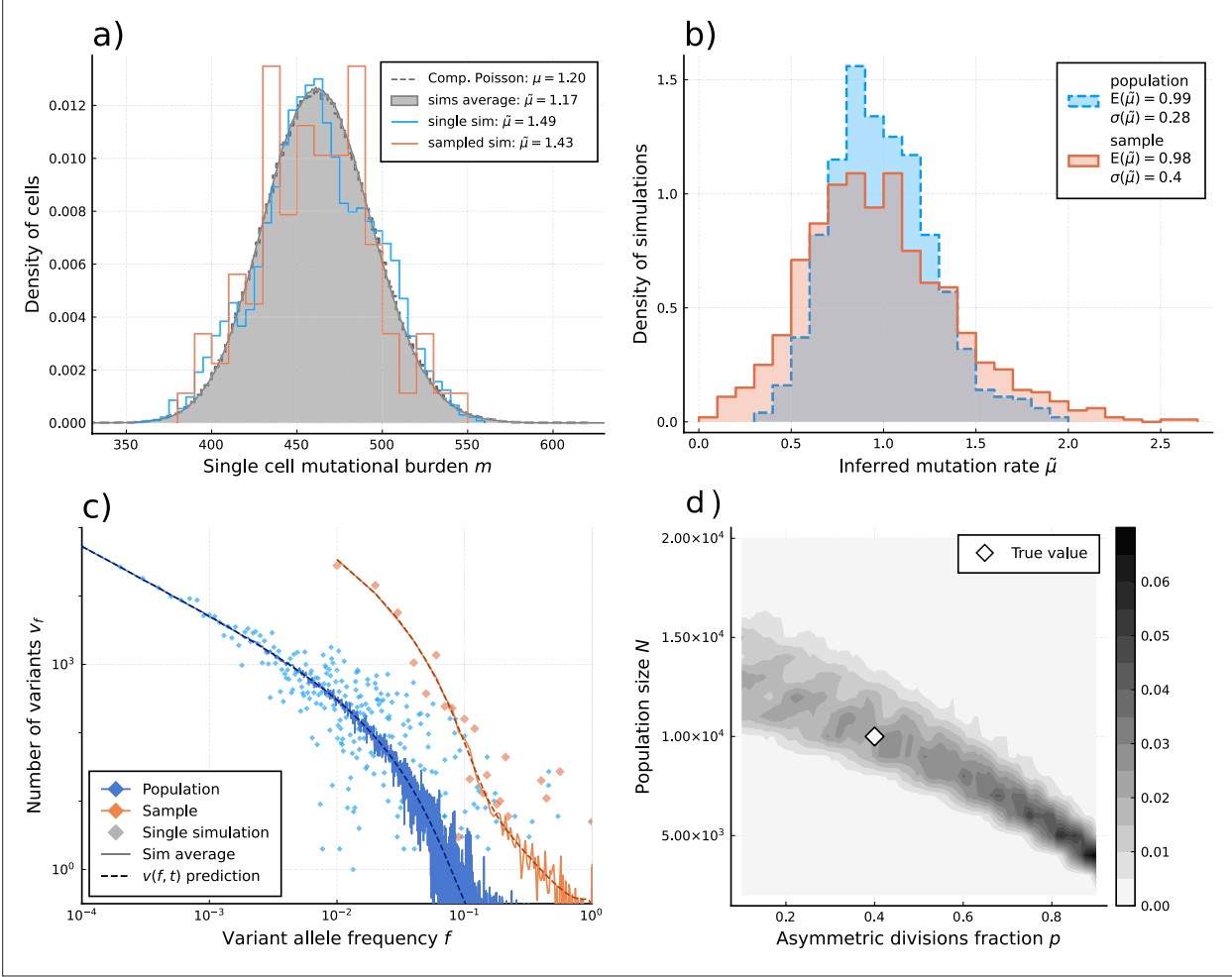

**Figure 3.** Inference of evolutionary parameters on simulated stem cell populations. Simulated populations were run up to age 59, growing exponentially from a single-cell to constant size $N_M = 10'000$ at age $t_M = 5$, with mutation rate $\mu = 1.2$ and division rates $\lambda = 5$ and $p = 0.4$. Where sampling is mentioned, the sample size 89 was taken. (**a**) The single-cell mutational burden distribution. The compound Poisson distribution (dashed line) matches the burden distribution when averaging over multiple independently evolved populations (filled curve). (**b**) Distribution of estimated mutation rates from 10'000 individual simulations, obtained from burden distributions of the complete populations (blue) as well as sampled sets of cells (orange). Because the expected mutational burden distribution is unaltered by sampling, the expected estimate of the mutation rate from *Equation 5* remains unchanged: $E(\tilde{\mu}_{pop}) = E(\tilde{\mu}_{sample})$. However, sampling increases the noise on the observed burden distribution, which results in a higher error margin of the estimate: $\sigma(\tilde{\mu}_{pop}) < \sigma(\tilde{\mu}_{sample})$. (**c**) VAF spectra measured in the complete population (blue) and a sampled set of cells (orange). In contrast with the mutational burden distribution, strong sampling changes the shape of the expected distribution. A single simulation result is shown (diamonds) alongside the theoretically predicted expected values for both the total and sampled populations (*Equations 1 and 12*) (dashed line) and the average across 100 simulations (solid line). (**d**) Distribution of $N_M$ and $p$ inference results for 100 simulated and sampled populations, through estimation of $\tilde{\mu}$ and $\tilde{\lambda}$ from the single-cell burden distribution and fitting the number of lowest frequency ($1/S$) mutations to the theoretical prediction in *Equation 1* (see *Figure 3—figure supplements 1–3* as well).

The online version of this article includes the following figure supplement(s) for figure 3:

**Figure supplement 1.** Stochastic simulations of the Variance/Mean of the mutational burden distribution over time for a per cell division mutation rate of $\mu = 1.3$ and varying stem cell population size $N$ and asymmetric division probability $p$.

**Figure supplement 2.** Likelihood of the Variance/Mean to be in the interval $3 < \theta < 5$ for a per division mutation rate of $\mu = 1.3$.

**Figure supplement 3.** Likelihood of the Variance/Mean to be in the interval $3 < \theta < 5$ for a per division mutation rate of $\mu = 3$.

In fact, the previous estimates of 1.2–1.3 mutations per genome per division were derived from early developmental cell divisions where the effects of aging would be negligible. The increased mutation rate inferred here could in part be explained by proliferation-independent effects. However, it is important to note that the true distribution of divisions per lineage may also be over-dispersed

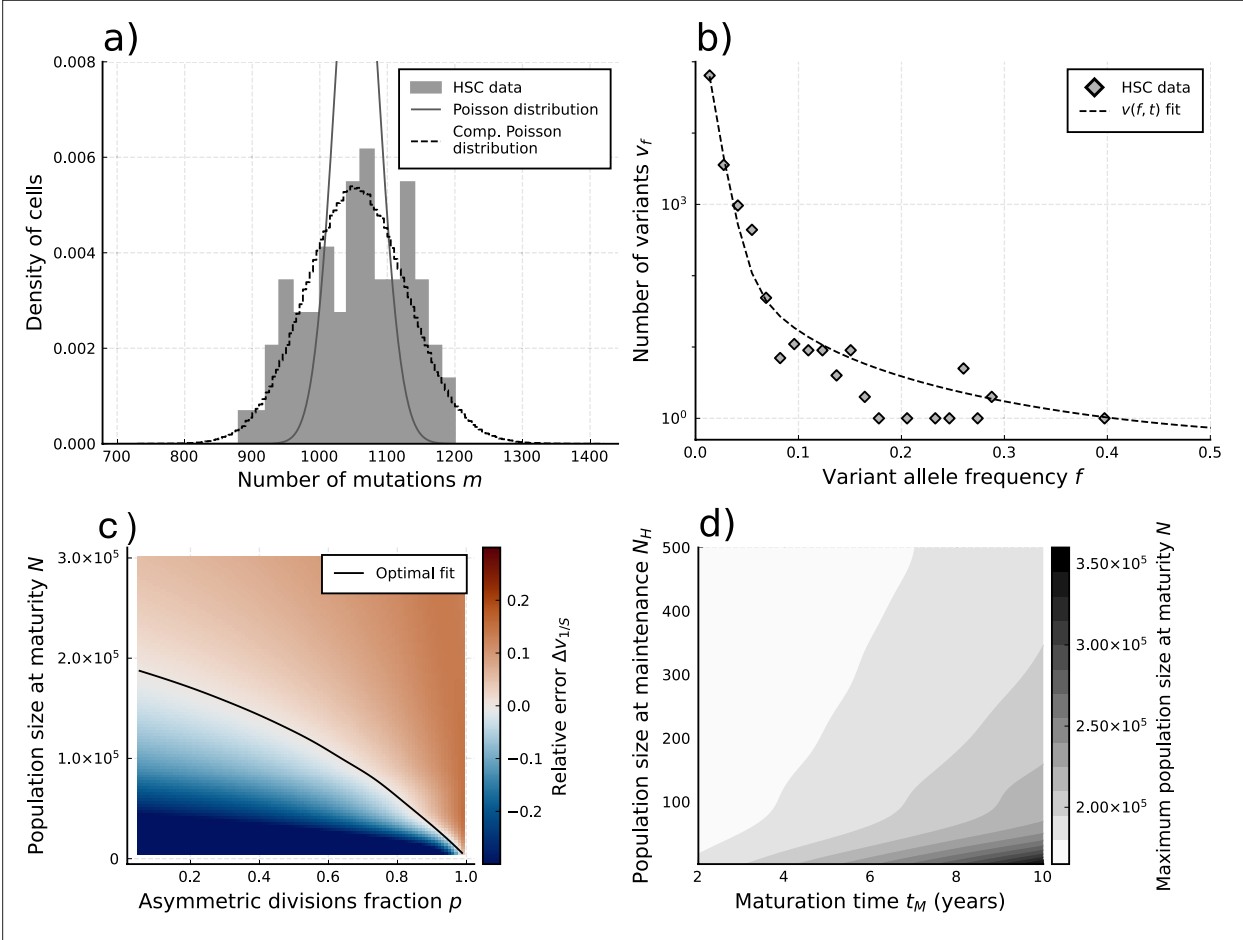

**Figure 4.** Evolutionary inferences in single-cell hematopoietic stem cell (HSC) data. (**a**) The single-cell mutational burden distribution of the data (bars) and the compound Poisson distribution obtained from its mean and variance, used to obtain the estimated per division mutation rate $\tilde{\mu}$. (**b**) Distribution of mutation frequencies of the data and theoretically predicted average fitted to only the lowest frequency ($1/S$) data point. (**c**) Difference $\Delta v_f$ between the measured value of the VAF spectrum at the lowest frequency ($1/S$) and its prediction from **Equation 1**, for varying total population size $N$ and asymmetric division proportion $p$, with fixed maturation time $t_M = 5$ and operational hematopoietic population size $N_H = 50$. The solid line denotes the plane of best fit where this difference is 0. (**d**) Maximally inferred population size $N$ (taking $p = 0$ in (**c**)) for variation of the maturation time $t_N$ and the operational hematopoietic population size $N_H$ (see **Figure 4—figure supplement 1** as well).

The online version of this article includes the following figure supplement(s) for figure 4:

**Figure supplement 1.** The standard deviation on the variant allele frequency (VAF) spectrum increases for higher frequencies.

compared to the Poisson model (i.e. $\mathrm{Var}(y) > \mathrm{E}(y)$), which would lead to an overestimation of $\mu$ in **Equation 5**. Furthermore, the observed burden distribution is incompatible with a simple division-independent mutation model, which would lead to a Poisson distribution of the burdens that has a variance much smaller than what we observed (**Figure 4a**). A combination of division-independent mutations together with non-Poissonian cell divisions could possibly reproduce the result.

## Sparse sampling, single-cell derived VAF spectra, and evolutionary inferences

Current methods limit us to whole genome information of at most a few thousand single cells per tissue or tumor (**Lim et al., 2020**). In many situations, this will only constitute a small fraction of the underlying long-term proliferating cell population. For example, although the exact number of HSCs remains unknown, most recent estimates suggest a population size of $10^5$ cells (**Watson et al., 2020**; **Lee-Six et al., 2018**). In tumors, this number can be as large as $10^{10}$ cells (**Werner et al., 2016**). As we have shown in the previous paragraph, the single-cell mutational burden distribution allows certain

inferences even if sampling is sparse. If we, however, construct the VAF spectrum from such a small sample, the distribution of variants will be significantly transformed with respect to that of the total population (*Figure 3c*). Since *Equation 1* gives the expected curve measured from the total population, one must include a correction, which is given by the expectation of hypergeometric sampling (i.e. without replacement). This can be obtained through the transformation (see Sampling affects the observed VAF distribution).

$$\tilde{V}(i) = \sum_{j=i}^{N} V(j) \cdot \binom{j}{i} \binom{N-j}{S-i} \bigg/ \binom{N}{S} \tag{6}$$

with $\tilde{V}$ the VAFs observed in the sample and $S$ the sample size. Furthermore, from stochastic simulations we note that the variance in the distribution increases with variant frequency $f$, making the lowest frequency state (i.e. $1/S$) the best candidate for comparative fitting and evolutionary inferences (see *Figure 4—figure supplement 1*).

Taking the mutation rate from *Werner et al., 2020* and the total division rate obtained from the burden distribution as constant, we explored solutions of *Equation 1* (sampled by *Equation 12*) for a wide range of realistic values in the remaining parameter space $\{N, p, t_M, N_H\}$. Comparing the lowest sampled frequency state ($f = 1/89$) with the data (*Figure 4b*) we identify a curve in the space of the numbers of HSCs $N$ and $p$ (fraction of divisions that are asymmetric) where theoretically predicted VAF distributions are identical (*Figure 4c*). Interestingly, this curve in $N$-$p$ space naturally identifies a maximal stem cell population size capable of producing the data, which occurs when stem cells divide exclusively symmetrically ($p = 0$). Variations of $t_M$ (the time until maturity of the HSC population) and $N_H$ (the HSC pool size at which maintenance begins to occur) had only a small effect on this inference (*Figure 4d*). This is partly because mutations arising during these phases are more likely to be found at higher frequencies upon measurement (*Poon et al., 2021*; *Werner, 2021*). We find the data to be congruent with an adult HSC pool of at most 200'000–300'000 cells, depending on the exact timings of the maturation phase (*Figure 4d*). This estimate agrees with the original study and other independent inferences based on population data (*Watson et al., 2020*; *Lee-Six et al., 2018*). However, this upper bound corresponds to the case of exclusively symmetric stem cell divisions. Accounting for the possibility of mutating asymmetric divisions reduces the estimated stem cell pool. For example, the data would be consistent with 50'000 stem cells if 90% ($p = 0.9$) of all stem cell divisions are asymmetric. In principle, even smaller stem cell pool sizes are possible, though decreasing an order of magnitude would imply an extremely large portion ($gt_{99}$%) of asymmetric HSC divisions. We note that in the data non-developmental branchings are observed, which immediately implies that not all HSC divisions can be asymmetric ($p < 1$) and thus a scenario of an extremely small population of only a handful of HSCs maintaining hematopoisis can be rejected. With an orthogonal population-based method, Watson and colleagues estimated 25'000 as a reasonable lower bound for the number of HSCs. Based on our inference, this would imply 0.95 as an upper bound for the proportion of all stem cell divisions that are asymmetric.

## Discussion

Here, we have shown that single-cell and bulk whole genome sequencing can inform on different aspects of somatic evolution. Single-cell information does support previous bulk-based estimates of possible ranges of HSC numbers. Surprisingly, the same single-cell data leads to a mutation rate inference during homeostasis is approximately four times higher compared to previous developmental estimates.

Another open question is the role of selection and how it shapes intra-tissue genetic heterogeneity. Evidence is emerging that positively selected variants in blood are almost universally present in individuals above 60, while the effective observable dynamics in younger individuals is well described by neutral dynamics. How results presented here generalize or modify will critically depend on the model of selection realized in human hematopoiesis, e.g., a model of rare or frequent driver events. Details of the underlying biology are currently unknown.

An important recent study has suggested an amended model of mutation accumulation in somatic tissues, wherein the majority of mutations are acquired continuously over time throughout a cell's

lifespan rather than during DNA replication (*Abascal et al., 2021*). We note that for a time-dependent rate of mutation accumulation that is constant across all cells at a given point in time, this model predicts Poisson distributed single-cell mutational burdens. In contrast, we find the single-cell mutational burden distribution in human HSCs to be highly over-dispersed compared to a Poisson model. A time-associated model of mutation accumulation with a constant rate alone cannot generate such a pattern. At least one additional source of stochasticity would be needed. This could be fluctuations in the time dependent mutation rates, non-constant cell proliferation rates, e.g., bursts of stem cell divisions potentially in reaction to injury or disease, or a combination of both effects. Either way, our observations suggest that our current theoretical models of somatic evolution in healthy tissues are incomplete. There is evidence for unknown processes contributing to intra-tissue genetic heterogeneity. The precise nature of these processes remains an open question.

# Materials and methods
## Model system
We model a dynamical population of $N(t)$ cells, each able to carry multiple mutations. It evolves by the occurrence of divisions — which replace cells in the pool with their (potentially mutated) daughter cells — and removals — which can correspond to differentiation or any form of cell inactivation. As outlined in *Figure 1*, we model different types of divisions affecting the system:

i.   A symmetric (non-differentiating) division replaces a cell in the population with two daughter cells, each of which maintains the mutational profile of its parent and acquires a Poisson distributed number of new mutations.
ii.  A symmetric differentiation removes a cell from the population, as both of its daughter cells have differentiated.
iii. an asymmetric division replaces a cell with a single daughter cell — as the other daughter has differentiated — and also acquires a Poisson distributed number of mutations.

The rates at which these divisions occur depends on what stage of development the population is in. A simplest approach is to consider two broad phases of evolution: an initial growth phase where $N(t)$ increases according to some growth function, and a mature phase at a time $t_M$ where $N(t)$ is constant and stem cell divisions occur only to maintain homeostasis of the population. While the earliest stage of population expansion can reasonably be approximated as purely developmental — i.e., where little to no turnover occurs in the population — maintenance of the functional population (e.g. in the blood system) will at some point be required even as the stem cell pool continues to grow. Thus we also include a mixed phase occurring before maturity but only after the stem cell pool has reached a size $N_H$, where stem cells can differentiate and homeostasis must be maintained. Note that this size is expected to be relatively small, as the earliest hematopoietic cells appear 16 days post-conception (*Ivanovs et al., 2017*), and peripheral blood is already driven mostly by functional HSC's by 14 weeks post conception (*Spencer Chapman et al., 2021*). Constancy of the population size in the mature phase is ensured by applying a Moran model of divisions (*Ewens, 2004a*), where each non-differentiating symmetric division — which results in two daughter cells with the HSC phenotype — is simultaneously accompanied by a removal, either through a symmetric differentiation or some form of cell inactivation. Thus in our model, we distinguish three types of events which occur independently with exponentially distributed waiting times:

1. Growth divisions, which are non-differentiating symmetric divisions (i) occurring at rate $\gamma$, and cause the population to increase in size by one cell;
2. Moran events, which consist of the simultaneous occurrence of a non-differentiating symmetric division (i) (which causes the population to increase in size by 1) and a cell removal (ii) (which causes the population to decrease in size by 1) either through symmetric differentiation or inactivation. These events occur at rate $\rho$;
3. Asymmetric divisions (iii), in which one of the daughter cells is differentiated and the other is not, occurring at rate $\phi$.

In the early developmental phase there are only growth divisions; during the mixed phase, growth divisions occur alongside Moran events and asymmetric divisions; and in the mature phase, only the Moran and asymmetric division events occur. Note that under these dynamics the rates of the

**Table 1.** Evolutionary parameters appearing in the model system.

| Symbol | Description | Units |
|--------|-------------|-------|
| $N_M$ | Carrying capacity of the mature population | |
| $t_M$ | Age when the cell population reaches mature size | years |
| $N_H$ | Population size at homeostatic divisions(start of the mixed-growth phase) | |
| $\gamma$ | Symmetric division rates in early developmental phase | /year |
| $\rho$ | Symmetric division rate in homeostatic state | /year |
| $\Phi$ | Asymmetric division rate in homeostatic state | /year |
| $\mu$ | Mutation rate | /division/daughter |

different types of divisions changes across phases: the rate of non-differentiating symmetric divisions (sometimes referred to as self-renewals) is given by $\gamma$ during early development, $\gamma + \rho$ during mixed growth, and finally $\rho$ at maturity; while the rate of asymmetric divisions respectively jumps from 0 to $\Phi$ in the mixed phase, and remains at $\phi$ in the mature phase. This can be summarized as:

$$\gamma(t) = \gamma\,\Theta(t < t_M)$$
$$\rho(t) = \rho\,\Theta(N(t) > N_H) \tag{7}$$
$$\phi(t) = \phi\,\Theta(N(t) > N_H)$$

We assume exponential growth (see section 1.4 for discussion of other possible growth curves) from the stem cell population's most recent common ancestor (the zygote, or any later occurring stem cell), and make the simplifying assumption that all division rates remain constant in their respective range of operation. The population size then depends on the division rates as $N(t) = \exp(\gamma(t)t)$ with $\Theta(x)$ the Heaviside step function.

We introduce $\mu$ as the rate parameter of the Poisson distribution from which the number of mutations arising in a daughter cell is drawn, i.e., the mutation rate per daughter per division. Each new mutation is presumed to be distinct from all previously existing mutations in the system, an assumption known as the *infinite sites model*.

A summary of all parameters governing this model is given in *Table 1*.

## Stochastic simulation of the dynamical model

In direct simulations of the system we model individually distinct cells, each carrying a particular set of mutations. These are evolved according to the model described in Section 1.1 by means of a Gillespie algorithm (*Gillespie, 1977*): Events are performed successively with exponentially distributed waiting times, and the acting cell in each event is selected uniformly randomly from the population. Daughter cells arising from a division inherit all mutations from their parent cell, and acquire new mutations which are distinct from any existing mutations.

Simulations are initiated with a single-cell carrying no mutations. In the early and mixed growth phases, taking the rate of growth divisions $\gamma$ as fixed ensures exponential growth of the complete population. To investigate the effect of differing growth curves (see section 1.4), we also implemented a logistic growth phase, where cell division gradually decreases and cell death gradually increases until the population reaches its mature size. The average population growth then follows

$$N(t) = \frac{N_M}{1 + e^{-\gamma t}}. \tag{8}$$

## The expected VAF distribution of the complete population

To obtain the dynamics of the VAF distribution, we first look at the probabilistic time evolution of a single variant's size (i.e. the number of cells sharing that mutation) $P_k(t)$, which is given by a Fokker-Plank equation. We first sketch the procedure for obtaining the result for a Moran model, where the population size is constant. Next, we show how to obtain the expression for a deterministically growing population. Finally, we use these results to obtain the distribution of variants for a population that changes in size.

## Probability density distribution of a single variant in a Moran process

The discrete probability distribution of the variant's size is governed by the transition probabilities $\mathbb{P}\{l|k\}$, which denotes the likelihood of the number of cells carrying this mutation changing from $k$ to $l$. In a Moran model, a constant population size $N$ is maintained by the simultaneous occurrence of one cell division and one cell death/removal. Thus, the possible transitions following the occurrence of such an event are given by the probabilities

$$\begin{cases} \mathbb{P}\{k+1 \,|\, k\} = \frac{k}{N}(1 - \frac{k}{N}) = T_{k+1,k} = T_k, \\[2mm] \mathbb{P}\{k-1 \,|\, k\} = \frac{k}{N}(1 - \frac{k}{N}) = T_{k-1,k} = T_k, \text{ and} \\[2mm] \mathbb{P}\{k \,|\, k\} = 1 - (T_{k+1,k} + T_{k-1,k}). \end{cases} \tag{9}$$

Assuming events occur with exponentially distributed waiting times at rate $\rho$, $P_k$ can be described through the master equation

$$\frac{1}{\rho}\frac{dP_k}{dt} = T_{k-1}P_{k-1} + -2T_k P_k + T_{k+1}P_{k+1} \tag{10}$$

When the population size $N$ is large, this leads to a system of $N$ differential equations, which is computational taxing to solve. Instead, a diffusion approximation can be made (on the assumption that $N$ is large, i.e., that there are many states $k = 0, 1, \ldots, N$), to describe $p(\kappa)$, the probability density of variant's size on the continuous support $\kappa \in [0, N]$. To this end, a Fokker-Planck equation is constructed of the form

$$\partial_t p(\kappa, t) = -\partial_\kappa A(\kappa, t)p(\kappa, t) + \partial_\kappa^2 B(\kappa, t)p(\kappa, t)/2 \tag{11}$$

where the coefficients are given through the infinitesimal propagator $t(\kappa + \Delta\kappa, t + \Delta t \,|\, \kappa, t)$:

$$\begin{aligned} A(\kappa, t) &= \lim_{\Delta t \to 0} \frac{1}{\Delta t} \int d(\Delta\kappa)\Delta\kappa\, t(\kappa + \Delta\kappa, t + \Delta t \,|\, \kappa, t) \\ B(\kappa, t) &= \lim_{\Delta t \to 0} \frac{1}{\Delta t} \int d(\Delta\kappa)(\Delta\kappa)^2\, t(\kappa + \Delta\kappa, t + \Delta t \,|\, \kappa, t) \end{aligned} \tag{12}$$

The integrals are the first and second moments of a displacement $\Delta\kappa$, and can be obtained using the heuristic (***Ewens, 2004b***):

$$\begin{aligned} \langle \Delta\kappa \rangle_{\Delta t} &\approx \langle \Delta\kappa \rangle_{1/\rho}\, \Delta t + \vartheta(\Delta t^2) = \sum_{\Delta k} \Delta k\, T_{k+\Delta k, k}\, \rho\, \Delta t + \vartheta(\Delta t^2) \\ \left\langle \Delta\kappa^2 \right\rangle_{\Delta t} &\approx \left\langle \Delta\kappa^2 \right\rangle_{1/\rho}\, \Delta t + \vartheta(\Delta t^2) = \sum_{\Delta k}(\Delta k)^2\, T_{k+\Delta k, k}\, \rho\, \Delta t + \vartheta(\Delta t^2) \end{aligned} \tag{13}$$

with $_{\Delta t}$ the expectation taken over a time interval $\Delta t$. Typically a transformation to frequency space is first performed, $f = \rho/N$, where $f$ is the frequency of cells carrying this mutation in the population. Thus, we obtain

$$A(f, t) = 0, \quad B(f, t) = \frac{2\nu}{N^2}f(1 - f) \tag{14}$$

and finally the well-known result

$$\partial_t p(f, t) = (\rho/N^2)\, \partial_f^2 f(1 - f)p(f, t) \tag{15}$$

## Probability density distribution of a single variant in a deterministic growing population

We now consider a growing population due to additional divisions occurring—at rate $\gamma$ per cell—alongside the previously described Moran events. For simplicity, we assume the population size $N(t)$ increases deterministically according to the growth rate. The transition probabilities for the variant sizes are then given by

$$\begin{cases} \mathbb{P}\{k+1, t+\Delta t \,|\, k, t\} = \dfrac{k}{N(t)}(1 - \dfrac{k}{N(t)})\rho N(t)\Delta t + \dfrac{k}{N(t)}\gamma N(t)\Delta t, \\[2mm] \mathbb{P}\{k-1, t+\Delta t \,|\, k, t\} = \dfrac{k}{N(t)}(1 - \dfrac{k}{N(t)})\rho N(t)\Delta t, \text{ and} \\[2mm] \mathbb{P}\{k, t+\Delta t \,|\, k, t\} = 1 - \mathbb{P}\{k+1, t+\Delta t \,|\, k, t\} - \mathbb{P}\{k-1, t+\Delta t \,|\, k, t\}. \end{cases} \qquad (16)$$

Using the previously described heuristic (19) the coefficients of the Fokker-Planck equation become

$$A(\kappa, t) = \kappa\gamma$$
$$B(\kappa, t) = 2\kappa[1 - \kappa/N]\rho + \kappa\gamma \qquad (17)$$

And the full equation

$$\partial_t p(\kappa, t) = -\partial_\kappa \gamma\kappa\, p(\kappa, t) + \partial_\kappa^2 [\kappa(1 - \kappa/N(t))\rho + \kappa\gamma/2]\, p(\kappa, t). \qquad (18)$$

## The expected VAF distribution in a varying population

For independently evolving variants, we denote the number of variants at size $k$ (i.e. shared by $k$ cells) as $V_k(t)$. We now note that in the limit of infinite variants, the fluxes between adjacent $k$-states are identical to the above-derived transition probabilities. Thus, the above-derived equations can similarly describe the expected dynamics of an ensemble of variant sizes $V(\kappa, t)$, where we once again introduce the parameter $\kappa \in [0, N(t)]$ to represent the continuous space analog of the variant size $k$. In our system of interest we, however, also wish to include an injected flux of newly arising mutations, which is limited to the state $\kappa = 1$ and has amplitude

$$\mathcal{C} = 2\mu(\rho + \gamma + \phi/2) \qquad (19)$$

Note that because asymmetric divisions result in just a single daughter remaining in the system, they contribute only half as many novel mutations to the system as symmetric divisions. In the continuous domain we introduce this flux as a Dirac delta function $\mathcal{C}\delta(\kappa - 1)$. Note also that, while these diffusion equations are more commonly applied in the frequency domain $f \in [0, 1]$, for the purpose of numerical simulation it will be convenient to remain in the space of absolute sizes $\kappa = fN(t)$. Then finally, the time evolution of the VAF distribution is given by

$$\partial_t V(\kappa, t) = -\partial_\kappa \gamma\kappa V(\kappa, t) + \partial_\kappa^2 [\kappa(1 - \kappa/N(t))\rho + \gamma\kappa/2]V(\kappa, t) + 2\mu N(t)(\rho + \gamma + \phi/2)\delta(\kappa - 1), \qquad (20)$$

which is the result stated in the main text.

## Numerical approximation of the VAF distribution dynamics

While a solution to the fixed population Moran system (21) is known (**Kimura, 1955**), it is unwieldy, and does not readily generalize to the more complex expressions (24) and (26). We, therefore, resorted to numerical approximations of their solution. We applied a method of lines approach with a finite difference discretization in the size coordinate $\kappa$. The incoming flux of variants at $\kappa = 1$ introduces a discontinuity in the derivative, which leads to stiffness in the system. This is the reason we eschewed the transformation to $f = \kappa/N(t)$, since when working in frequency space the point of incoming flux would change position as the population grows. To maximize performance near the discontinuity, we implemented a variable step size smallest near $\kappa = 1$, with the distances in space given by

$$\Delta\kappa_i = 1 + (i - 1) \cdot \alpha_i \qquad (21)$$

where $i \in 1, \ldots, n - 1$, with $n$ the number of discretized points, and

$$\alpha_i = 2\frac{N - (n - 1)}{(n - 1)(n - 2)} \qquad (22)$$

In this formalism the delta function in (1) was approximated as a step function with height $2/(\Delta\kappa_1 + \Delta\kappa_2)$.

**Table 2.** Evolutionary parameters appearing in the analytical derivations of the expected VAF distribution in the Moran and pure-birth models.

| Symbol | Description |
|---|---|
| $N$ | Total number of cells |
| $k$ | Abundance or number of cells < N |
| $V_k(t)$ | Number of mutations with abundance k |
| $P_k$ | Likelihood for a mutation with abundance k to increase or decrease |
| $C_k(t)$ | The average number of mutations with abundance k that increase or decrease |
| $\frac{dV_k(t)}{dt}$ | Average change of $V_k(t)$ per time step |
| $\mu$ | Mutation rate per daughter cell |

## Changing the growth function during the maturation phase

In the main text we have shown that, for a constant population, the past occurrence of a growth phase influences the shape of the observed VAF spectrum. Here, we investigate to what extent the precise form of this growth function alters the observation. To this end, we compare the expected VAF spectrum, calculated from *Equation 1*, for two different growth models: One where the population grows exponentially until the maximal capacity $N_M$ is reached, at which point it immediately stops expanding; and one where the population grows according to the logistic function, which is a common model for population growth in ecological systems. Since in the latter case the function only asymptotically approaches capacity, we take maturity to be the point where the population reaches $0.99 N_M$. The resulting growth functions and expected VAF spectra are shown in *Figure 2—figure supplement 1* for two time points measured after maturity. We observe that there is indeed a quantitative difference between the spectra of the two models. However, we note the same qualitative characteristics as described in the Results section of the main text: a pronounced move towards the fixed size shape in the lower frequencies, and a delayed response at higher frequency. The difference between the two models furthermore diminishes as the measurement occurs farther away from the end of the growth phase.

## Analytical solutions for the expected VAF distribution in select cases

While in section 1.3 it was shown how the expected VAF distribution of our stochastic model can be calculated numerically, under certain (stricter) conditions some solutions can also be obtained analytically. We present here derivations for analytical solutions of the distribution in a constant population adhering to a Moran process, and a strictly growing population subject to the pure-birth process. Although the results derived in this section are themselves not novel (see e.g. *Durrett, 2015*; *Williams et al., 2016*, and *Gunnarsson et al., 2021*), we present them here for completeness, with new derivations that facilitate a unified analysis and do not require readings of the previous literature. For convenience, a list of the parameters used in this section is given in *Table 2*.

### The equilibrium VAF distribution in a Moran process

In a birth-death Moran process, one cell dies and one is born every time step. Then, let every birth coincide with an average of $\mu$ new unique neutral mutations. Every cell can hold any number of mutations, and since these are neutral, the cells that are dying/born are chosen completely at random. If two cells are chosen for death and birth, any particular mutation has a chance $P_k$ to be in one of the cells but not in the other:

$$P_k = \frac{k}{N} \cdot (1 - \frac{k}{N})$$  (23)

This was previously also shown in *Equation 15*. If we want to know how the number of mutations changes, we need to know how many mutations increase or decrease in abundance on average. This simply means that we multiply $P_k$ by the number of mutations with the given abundance $k$:

$$C_k(t) = \frac{k}{N} \cdot (1 - \frac{k}{N}) \cdot V_k(t) \tag{24}$$

Note that because of the symmetry of the process, $C_k(t)$ is both the average increase and decrease. Lastly, if we look at a certain abundance $k$, there are two ways to lose mutations and two ways to gain mutations: $M_k$ increases if mutations of abundance $k + 1$ lose one cell or if mutations of abundance $k + 1$ gain one cell. $M_k$ decreases if mutations of abundance $k$ lose or gain a cell. Note that all of these things can happen at the same time. If we want to know the average change $\frac{dV_k(t)}{dt}$, we, therefore, need to incorporate all of these four possibilities. After we add the rate division rate $\rho$, we finally get the following equation for the average change per unit of time:

$$\frac{dV_k(t)}{dt} = \rho \cdot (C_{k-1}, t) + C_{k+1}(t) - 2C_k(t) \tag{25}$$

This is a variation of *Equation 16* applied to all mutations, as opposed to just a single one. For $k = 1$ the first term $C_0$ is simply replaced by a fixed mutation rate $\mu$, for $k = N$, the middle term has to be removed (and the last term $C_N(t)$ is always 0). Once we apply this to an entire distribution of mutations, it gives us an estimate of how, on average, this distribution changes every time step. Notably, it should approach some equilibrium state if we exclude the last state, which is absorbing. Let us consider the distribution from $k = 1$ to $k = N - 1$. The only way for this distribution to lose mutations is by either a mutation of abundance 1 to die out, or a mutation of abundance $N - 1$ to fixate. This implies an equilibrium at:

$$\sum_{k=1}^{N-1} \frac{dV_k(t)}{dt} = \rho \cdot (\mu - C_{N-1}(t) - C_1(t)) = 0 \tag{26}$$

$$\implies \mu = C_1(t) + C_{N-1}(t) \implies C_1(t) = \mu - C_{N-1}(t)$$

We can now apply this iteratively to the other $C_k(t)$ (for convenience, we ignore the $\rho$, since it has no bearing on the equilibrium, merely on how fast we approach it):

$$\frac{dV_1(t)}{dt} = \mu + C_2(t) - 2C_1(t) = 0 \implies C_2(t) = 2(\mu - C_{N-1}(t)) - \mu = \mu - 2C_{N-1}(t)$$

$$0 = C_1(t) + C_3(t) - 2C_2(t) \implies C_3(t) = 2(\mu - 2C_{N-1}(t)) - \mu + C_{N-1}(t) = \mu - 3C_{N-1}(t) \tag{27}$$

$$\cdots$$

$$C_k(t) = \mu - kC_{N-1}(t)$$

Which can then be applied to $C_{N-1}(t)$ itself:

$$C_{N-1}(t) = \mu - (N-1)C_{N-1}(t) \implies C_{N-1}(t) = \frac{\mu}{N} \tag{28}$$

This can now be used to solve for $C_1(t)$, $C_1(t)$ … $C_{N-2}(t)$ iteratively, resulting in an equilibrium solution $C_k^*$:

$$C_k^* = \mu \cdot \frac{N - k}{N} \tag{29}$$

Finally, this can be transformed back into the $V_k(t)$ to get the equilibrium solution $V^*(k)$:

$$V^*(k) = C_k^* \cdot \frac{N}{k} \cdot \frac{1}{1 - \frac{k}{N}} = \mu \cdot \frac{N - k}{N} \cdot \frac{N}{k} \cdot \frac{N}{N - k} = \mu \frac{N}{k} \tag{30}$$

## The VAF distribution of a growing population under a pure-birth process

In a pure-birth process only births, and no deaths, occur in the population. Suppose then a strictly growing population starting from a single-cell increasing by 1 every time step, with a mutation rate of $\mu$. We denote the number of mutations with abundance k at time t as $V_k(t)$. Then we can set up a system for the average change per time step:

$$V_k(t+1) = (1 - \frac{k}{t+1})V_k(t) + \frac{k-1}{t+1}V_{k-1}(t)$$
$$V_1(t+1) = (1 - \frac{1}{t+1})V_1(t) + 2\mu$$

(31)

For simplicity we view the process in event time, so that every time step is exactly one division event. However, the process can be projected so that the amount of time $\delta t$ that passes between every event is drawn at random from a given distribution. Since this projection is always possible in both directions now matter the chosen distribution, this proof is valid for any process, as long as births are not allowed to be simultaneous. The $V_k(t)$ themselves are all strictly increasing every time step and hence approach infinity, but we can make a statement about the average shape of the distribution and the value of each specific points in time:

$$V_k(t) = (\frac{2}{k} - \frac{2}{k+1}) \cdot (t+1)\mu = \frac{2}{k \cdot (k+1)} \cdot (t+1)\mu$$

(32)

To prove that $2/(k \cdot (k+1))$ is indeed the expected distribution for this system, we will first show that the first several time steps follow this shape. Then, we prove that every time a new cell and thus a new possible abundance $k = t$ is added to the system, the new distribution at the new abundance will follow the expected distribution as well. Finally, we will show that if an abundance and the neighboring abundance follow the distribution already, it will continue to follow the distribution.

Let's show now that the first few time steps follow the claimed distribution:

$$V_1(1) = 2\mu$$
$$V_1(2) = (1 - \frac{1}{2}) \cdot 2\mu + 2\mu = 3\mu$$
$$V_2(2) = (\frac{1}{2}) \cdot 2\mu = 1\mu$$
$$\implies \frac{V_2(2)}{V_1(2)} = \frac{1}{3} = \frac{2}{2} - \frac{2}{3}$$
$$V_1(3) = (1 - \frac{1}{3}) \cdot 3\mu + 2\mu = 4\mu$$
$$V_2(3) = (1 - \frac{2}{3}) \cdot 3 + \frac{1}{3} \cdot 1\mu = \frac{4}{3}\mu$$
$$V_3(3) = (\frac{2}{3}) \cdot 1 = \frac{2}{3}\mu$$
$$\implies \frac{V_2(3)}{V_1(3)} = \frac{1}{3} = \frac{2}{2} - \frac{2}{3}$$
$$\implies \frac{V_3(3)}{V_1(3)} = \frac{1}{6} = \frac{2}{3} - \frac{2}{4}$$

(33)

Now, since the first steps seem to follow the distribution as expected, we can move on to showing that generally at time step $t$, the newly generated abundance $k = t$ will always fit the distribution:

$$V_t(t) = 0 + \frac{t-1}{t} \cdot \frac{2}{(t-1) \cdot t} \cdot t \cdot \mu = \frac{2}{t} \cdot \mu$$
$$= \frac{2}{t \cdot (t+1)} \cdot (t+1) \cdot \mu$$
$$= \frac{2}{k \cdot (k+1)} \cdot (t+1) \cdot \mu$$

(34)

The only part remaining is proving that if the preceding time steps fit the distribution, the following time step also follows the distribution:

$$
\begin{aligned}
\frac{V_k(t+1)}{V_1(t)} &= (1 - \frac{k}{t+1})\frac{V_k(t)}{V_1(t)} + \frac{k-1}{t+1}\frac{V_{k-1}(t)}{V_1(t)} = (1 - \frac{k}{t+1})(\frac{2}{k \cdot (k+1)}) + \frac{k-1}{t+1}(\frac{2}{(k-1) \cdot (k)}) \\
&= (1 - \frac{k}{t+1})(\frac{2}{k \cdot (k+1)}) + \frac{k+1}{t+1}(\frac{2}{(k+1) \cdot (k)}) = (1 - \frac{k}{t+1} + \frac{k+1}{t+1})(\frac{2}{k \cdot (k+1)}) \\
&= (1 + \frac{1}{t+1})(\frac{2}{k \cdot (k+1)}) \\
V_1(t+1) &= (1 - \frac{1}{t+1}) \cdot (t+1)\mu + 2\mu = (t+1-1+2)\mu = (t+2)\mu \\
&\implies \frac{V_1(t+1)}{V_1(t)} = \frac{t+2}{t+1} = 1 + \frac{1}{t+1} \\
&\implies \frac{V_k(t+1)}{V_1(t+1)} = \frac{2}{k \cdot (k+1)} \implies V_k(t+1) = \frac{2}{k \cdot (k+1)} \cdot (t+2)\mu
\end{aligned}
\tag{35}
$$

Note that unlike for the constant population case, the expected distribution does not need time to asymptotically approach the steady-state solution, as it is already at equilibrium from $t = 1$.

## Sampling affects the observed VAF distribution

*Equation 1* describes the evolution of mutation frequencies with respect to the whole population, however, these are typically not known from sequencing data. Instead, an observation provides the frequencies of mutations as measured in the sequenced sample of the population. This distinction should be recognized when interpreting experimental data, as the statistical distribution for the sample's VAF distribution differs from that of the true population (*Figure 3c*). To obtain the expected distribution $\tilde{v}(f)$ from a sample of $S$ cells, we must consider how each mutation's frequency in the true population can be altered in the sample, so that the resulting curve is given by the sum over all original mutation frequencies $jN$ weighted by their probability of being sampled to a different frequency $iS$. If sampling is performed randomly this probability is given by the hypergeometric distribution (sampling without replacement), leading to the expression

$$
\tilde{v}(i/S) = \sum_{j=1}^{N} v(j/N) \cdot \binom{j}{i}\binom{N-j}{S-i} / \binom{N}{S}
\tag{36}
$$

Besides the anticipated loss of resolution, the effect of this operation is to shift the expected mutation frequencies upward with respect to their true frequency in the population (see *Figure 3c* in the main text), which can lead to improper inferences if not correctly accounted for.

## The mutational burden distribution

### The expected mutational burden distribution

The expected distribution of mutational burdens can be obtained from the argument that the burden $m_j$ of an individual cell at the time of observation must equal the sum over all mutations that occurred during each of its past cell divisions:

$$
m_j = \sum_{i}^{y_j} u_{ji}
\tag{37}
$$

Without knowledge of the distributions of $y$ and $u$ other than the requirement that the $u_{ji}$ are identically distributed and independently drawn, it is straightforward to show that the expectation and variance of $m_j$ are given by $\mathrm{E}(m) = \mathrm{E}(y)\,\mathrm{E}(u)$ and $\mathrm{Var}(m) = \mathrm{E}(y)\,\mathrm{Var}(u) + \mathrm{E}(u)^2\,\mathrm{Var}(y)$ (Supplement). With the number of novel mutations per division as Poisson distributed, the average number of divisions per lineage — expressed in terms of an effective division rate $\lambda$ — can thus be estimated from a sample as

$$
\int \lambda(t)dt = \mathrm{E}(y) = \mathrm{E}(m)/\mu
\tag{38}
$$

If the mutation rate $\mu$ is not known, it can similarly be obtained from

$$
\mu = \left(\frac{\mathrm{Var}(m)}{\mathrm{E}(m)} - 1\right)\frac{\mathrm{E}(y)}{\mathrm{Var}(y)}
\tag{39}
$$

Note that while a common modeling practice is to take the number of divisions as Poisson distributed — i.e., assuming exponentially distributed times between divisions — the estimates (*Equation 4*) and (*Equation 5*) do not require this. In fact, the factor $E(y)/\text{Var}(y)$ in (*Equation 5*) can be interpreted as a measure for the under- or over-dispersion of divisions per lineage with respect to the Poisson model where $E(y) = \text{Var}(y)$. In other words, if the Poisson model is assumed (then (*Equation 37*) describes the compound Poisson distribution (CPD)) the mutation rate can be estimated directly from the data $m$, whereas if the mutation rate is known the accuracy of the Poisson model can be assessed. Finally, note that in the true system there are nonzero correlations between lineages – the early $u_{ji}$ are shared amongst many cells – meaning that for a single population the $m_j$ are not truly drawn from (*Equation 37*). While this does not affect the expected division rate (*Equation 4*), an inferred mutation rate will be influenced by a shared history, as it effectively reduces $\text{Var}(y)$ and $\text{Var}(u)$ (*Figure 3b*).

## Fluctuations of the mutational burden distribution

To investigate expected within- and between-patient variation of the mutational burden distribution we implemented individual-based stochastic simulations of single-cell mutation acquisition. We start simulations with $N$ cells and 0 mutations. Cells are picked randomly for division and acquire a number of mutations drawn from a Poisson distribution with rate parameter $\mu$. In addition, cells may undergo symmetric or asymmetric divisions with a given probability $p$. If a cell divides asymmetrically, the novel mutations are added to that cell in the simulation. If a cell divides symmetrically, one cell in the population replaces another and novel mutations are added to the replaced and the dividing cell.

These simulations allow us to track the mutational burden distribution and its variance for arbitrary $p$, $\mu$ and large $N$ over time. Realizations of these dynamics for $\mu = 1.3$ per cell division are shown in *Figure 3—figure supplement 2* and are compared to the experimental observation of the mutational burden distribution in *Durrett, 2013*. We observe significant stochastic variations, especially pronounced for small-stem cell population sizes $N$ and low asymmetric division probabilities $p$. However, the true biological parameters are more likely to be a combination of large $N$ and large $p$ and, therefore, a mutation rate of $\mu = 1.3$ per cell division is unlikely to explain the observations in *Lee-Six et al., 2018*.

In *Figure 3—figure supplement 2* and *Figure 3—figure supplement 3* we show the likelihood that the ratio of variance/mean is within the experimentally observed interval $3 < \theta < 5$. Again we can see that for $\mu = 1.3$ the ability to explain the experimental observations is limited for any combination of parameters. Increasing the mutation rate to $\mu = 3$ in contrast can explain the experimental observations for a wide range of parameter combinations.

## Data fitting pipeline

To estimate the values of the model parameters from the data, we combine both predictions for the single-cell mutational burden as well as the mutation allele frequency distribution. The unknown parameters are those listed in *Table 1*, whereby the growth rate is implicitly defined from the mature population size and the maturation time as $\gamma = \log(N_M)/t_M$.

## Estimating division rates from the single-cell mutational burden distribution

While the single-cell mutational burden distribution allows for direct estimation of the average number of divisions that occurred per lineage through *Equation 4*, this does not directly provide the division rates, as these combine in the integrand on the lhs. To perform the integration we must first determine the expected contributions of each type of division (growth, symmetric, or asymmetric) to the mutational burden of a single lineage, as they — perhaps counter-intuitively — do not influence the expected burden equally.

Consider the system of $N$ cells which obey the dynamics described in Section 1.1. At any point in time each cell describes a single lineage, and we are interested in the expected number of mutations added per lineage over time. Since divisions occur as independent Poisson events, we may treat the distinct types individually and simply add their contributions together to obtain the total burden increase over time. When an asymmetric division occurs the situation is simple: The single stem-type daughter cell acquires on average $\mu$ new mutations, which increase the average burden per cell by $\mu/N$. Since these dynamics are Markovian, and events occur at rate $N\phi$ in the population, we obtain the average number of mutations acquired per lineage in a time $\Delta t$ as

$$\Delta m_\phi = N\gamma \times (\mu/N) \times \Delta t = \phi\mu\,\Delta t \tag{40}$$

When a symmetric division occurs (as performed in the Moran process described previously) the situation becomes slightly more complicated. Following a single division we discern two disjoint outcomes: either two different cells are selected for replication and removal, or the same cell is selected for both. The former case occurs with probability $1 - 1/N$ and results in the average burden per lineage increasing by $2\mu/N$ (two daughter cells obtained on average $\mu$ new mutations), while the latter occurs with probability $1/N$ and causes the average burden to increase by $\mu/N$ (one of the daughter cells was removed). With these events occurring at rate $N\rho$ the expected increase per lineage after a time $\Delta t$ is:

$$\Delta m_\rho = N\rho\mu \left[ \left(1 - \frac{1}{N}\right) \frac{2}{N} + \frac{1}{N}\frac{1}{N} \right] \Delta t = \rho\mu(2 - 1/N)\Delta t \approx 2\rho\mu\,\Delta t \tag{41}$$

with the final approximation obtained in the limit of large $N$. For the pure growth divisions occurring at rate $N\gamma$ we take a similar approach, though we now consider an infinitesimal time increment $dt$ to account for the varying population size. After a single division, the average burden increase is $2\mu/(N(t) + 1)$, so that

$$dm_\gamma = N(t)\gamma\mu \frac{2}{N(t) + 1}dt = 2\gamma\mu \frac{N(t)}{N(t) + 1}dt \approx 2\gamma\mu\,dt \tag{42}$$

Thus, we find that the growth and Moran divisions on average add twice as many mutations to a lineage as the asymmetric divisions. Note that we obtain the average increase in lineage (the number of divisions in a cell's past) by setting $\mu = 1$ in the equations above. Plugging these results into the integral of (*Equation 4*), we find for the expected number of divisions per lineage at time.

$$y(t) = \int_0^t \left[ 2\rho \left(1 - \frac{1}{2N(\tau)}\right) + \phi + 2\gamma\left(1 - \frac{1}{N(\tau) + 1}\right) \right] d\tau \tag{43}$$

Taking into account the different phases of evolution we can separate the integral into three parts.

$$\begin{aligned}
y(t) = & \int_0^{\log N_H/\gamma} 2\gamma\left(1 - \frac{1}{N(\tau) + 1}\right) d\tau \\
& + \int_{\log N_H/\gamma}^{t_M} \left[ 2\rho\left(1 - \frac{1}{2N(\tau)}\right) + \phi + 2\gamma\left(1 - \frac{1}{N(\tau) + 1}\right) \right] d\tau \\
& + \int_{t_m}^t \left[ 2\rho\left(1 - \frac{1}{2N(\tau)}\right) + \phi \right] d\tau
\end{aligned} \tag{44}$$

Denoting $\lambda = \rho + \phi$ the total rate of homeostasis divisions within the stem cell compartment, performing the integration we solve this for,

$$\lambda = \frac{R - 2\log\left(N_M + 1\right) + \log(4)}{-(1 - p)(t - t_M)/N_M + (2 - p)\left(t - t_M\frac{\log(N_H)}{\log(N_M)}\right) - (1 - p)\,t_M\left(\frac{1}{N_H} - \frac{1}{N_M}\right)/\log(N_M)} \tag{45}$$

with $R = \mathrm{E}(m)/\mu$ the average number of divisions per lineage estimated from the mutational burden distribution. With $\mu$ estimated from *Werner et al., 2020*, the remaining unknown parameters are $t_M$, $N_H$, $N_M$, and $p$. Finding the standard error on $\mathrm{E}(m)$ as $\sigma_{\mathrm{E}(m)} = \sigma_m/\sqrt{S}$ (with $S$ as the sample size) and sweeping the unknown parameters within reasonable limits ($t_M \in [0, 10]$, $N_H \in [0, 1000]$, $N_M \in [20'000, 500'000]$) gives an error interval for the expected $\lambda$.

## Fitting to the lowest frequency of the VAF distribution

We must be cautious about attempting to directly fit realizations of *Equation 1* to an observed distribution in its entirety. Without knowledge of the variance across individual realizations for each possible allele frequency, there is no rigorous way to weigh the errors from different frequencies to obtain a single best fit. Indeed, simulations suggest that the variance in the number of mutations per VAF grows with the frequency (*Figure 4—figure supplement 1*), implying that a non-weighted fit would overemphasize fluctuations at one end of the spectrum. We, therefore, selected only the lowest

frequency available in the sample – which is anticipated to be the best described by the expected value (**Figure 4—figure supplement 1b**) – as the reference for parameter fitting.

For select values of $t_M$ and $N_H$, we calculated the VAF distribution — by numerically solving **Equation 1** and applying the sampling operator **Equation 12** — at lattice points in the parameter space of $N_M \in \left[5 \cdot 10^3, 5 \cdot 10^5\right] \times p \in [0, 1]$. For each point $(N_{Mi}, p_i)$ we compared the lowest frequency of the theoretical expected distribution $v_{1/S}$ with the value measured in the data, resulting in a 2D lattice of errors. From this lattice a continuous error function was obtained through spline interpolation, wherein we identified a zero-error line, which is the line of optimal fit shown in **Figure 4c**.

## Code availability

The code for the simulations was written in the Julia computing language Julia 1.7.3 or C++ (gnu++ 14), and can be accessed at https://github.com/natevmp/hsc-vaf-dynamics.

## Reused data

We have reused the publicly available data in **Lee-Six et al., 2018**, where the reused data in our **Figure 4** can be accessed at https://data.mendeley.com/datasets/yzjw2stk7f/1, and in **Martincorena et al., 2018**, where the reused data in our **Figure 2** can be accessed in the Resources section aau3879 tables2.xlsx at https://www.science.org/doi/10.1126/science.aau3879.

## Acknowledgements

The authors are grateful to Marc Williams and Christo Morison for discussions at different stages of the manuscript. NVMP acknowledges support by Télévie and the Barts Charity. BW is supported by a Barts Charity Lectureship (grant no. MGU045) and a UKRI Future Leaders Fellowship (grant no. MR/V02342X/1). This project has been facilitated by an Alan Turing Institute Data Science for Science Program grant TU/ASG/R-SPES-121.

## Additional information

### Funding

| Funder | Grant reference number | Author |
| --- | --- | --- |
| Barts Charity | MGU045 | Nathaniel V Mon Père |
| UK Research and Innovation | MR/V02342X/1 | Benjamin Werner |

The funders had no role in study design, data collection and interpretation, or the decision to submit the work for publication.

### Author contributions

Marius E Moeller, Conceptualization, Formal analysis, Validation, Methodology, Writing – original draft, Writing – review and editing; Nathaniel V Mon Père, Conceptualization, Formal analysis, Investigation, Visualization, Methodology, Writing – original draft, Writing – review and editing; Benjamin Werner, Conceptualization, Supervision, Funding acquisition, Investigation, Methodology, Writing – original draft, Writing – review and editing; Weini Huang, Conceptualization, Formal analysis, Supervision, Funding acquisition, Methodology, Writing – original draft, Writing – review and editing

### Author ORCIDs

Nathaniel V Mon Père ⓘ https://orcid.org/0000-0003-3561-2050
Benjamin Werner ⓘ http://orcid.org/0000-0002-6857-8699
Weini Huang ⓘ http://orcid.org/0000-0002-9016-2665

Reviewer #1 (Public Review): https://doi.org/10.7554/eLife.89780.3.sa1
Reviewer #2 (Public Review): https://doi.org/10.7554/eLife.89780.3.sa2
Author Response https://doi.org/10.7554/eLife.89780.3.sa3

# Additional files

## Supplementary files
• MDAR checklist

## Data availability
The current manuscript is a computational study, so no data have been generated for this manuscript. We have reused the publicly available data in *Lee-Six et al., 2018*, where the reused data in our Figure 4 can be accessed at https://data.mendeley.com/datasets/yzjw2stk7f/1, and in *Martincorena et al., 2018*, where the reused data in our Figure 2 can be accessed in the Resources section aau3879 tables2.xlsx at https://www.science.org/doi/10.1126/science.aau3879. The code for the simulations was written in the Julia computing language Julia 1.7.3 or C++ (gnu++ 14), and can be accessed at https://github.com/natevmp/hsc-vaf-dynamics (copy archived at *Mon Père and Moeller, 2023*).

The following previously published datasets were used:

| Author(s) | Year | Dataset title | Dataset URL | Database and Identifier |
|---|---|---|---|---|
| Martincorena et al. | 2018 | The evolving mutational landscape of normal human esophagus | https://web2.ega-archive.org/datasets/EGAD00001004158 | European Genome-Phenome Archive, EGAD00001004158 |
| Martincorena et al. | 2018 | The evolving mutational landscape of normal human esophagus | https://web2.ega-archive.org/datasets/EGAD00001004159 | European Genome-Phenome Archive, EGAD00001004159 |
| Lee-Six et al. | 2018 | The Causes of Clonal Blood Cell Disorders Study - SCOR (2018-04-19) | https://ega-archive.org/datasets/EGAD00001004086 | European Genome-Phenome Archive, EGAD00001004086 |
| Lee-Six H, Campbell P | 2018 | Population dynamics of normal human blood inferred from spontaneous somatic mutations - substitution calls | https://doi.org/10.17632/yzjw2stk7f.1 | Mendeley Data, 10.17632/yzjw2stk7f.1 |

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
