## [Editor Report · eLife assessment]

In this paper, the authors introduce **fundamental** work on mathematical methods for inferring evolutionary parameters of interest from RNA data in healthy tissue and during hematopoiesis. By combining single cell and bulk sequencing analyses, the authors use a stochastic process to inform different aspects of genetic heterogeneity; the strength of evidence in support of the authors' claim is **exceptional**. The work will be of broad interest to cell biologists and theoretical biologists.

---

## [Referee Report · Reviewer #1 (Public Review)]

Authors propose mathematical methods for inferring evolutionary parameters of interest from bulk/single cell sequencing data in healthy tissue and hematopoiesis. Authors attempt to go beyond previous models by including three phases of human development: early development, growth and maintenance, and mature phase. Introductory figures (1 and 2) provide the connection to previous analytical results (based on power laws), while figure 3 denotes the role of sampling effects, and figure 4 provides a real-world example.

This approach dovetails nicely with previous literature, providing clear insight into when previous theoretical results are valid and when they break down. Much of the previous literature is devoted to bulk sequencing, leading the authors to investigate the role of (sub)-sampling due to single cell data, where mutation burden and mutation rate distributions are easily recapitulated. Although not strongly emphasized in the manuscript, sub-sampling does increase noise leading to differences between population and sample distributions. From my view, these results provide an important contribution to the literature and are able to nicely describe and make inferences in a single cell HSC data set.

---

## [Referee Report · Reviewer #2 (Public Review)]

Summary: The authors provide a nice summary on the possibility to study genetic heterogeneity and how to measure the dynamics of stem cells. By combining single cell and bulk sequencing analyses, they aim to use a stochastic process and inform on different aspects of genetic heterogeneity.

Strengths: Well designed study and strong methods.

---

## [Author Response]

The following is the authors’ response to the original reviews.

We are very grateful for your time and efforts spent on our manuscript. Your feedback has been very valuable. Please see below a point-by-point response to each suggestion and actions taken to address each point in the manuscript.

**eLife assessment**
In this fundamental study, the authors propose analytical methods for inferring evolutionary parameters of interest from sequencing data in healthy tissue relevant to hematopoiesis. By combining analyses of single cell and bulk sequencing data, the authors can use a stochastic process to inform different aspects of genetic heterogeneity. The strength of evidence in support of the authors' claim is thus compelling. The work will be of broad interest to cell biologists and theoretical biologists.
**Public Reviews:**

**Reviewer #1 (Public Review):**
Summary:Authors propose mathematical methods for inferring evolutionary parameters of interest from bulk/single cell sequencing data in healthy tissue and hematopoiesis. In general, the introduction is well-written and adequately references the relevant and important previous literature and findings in this field (e.g. the power laws for well-mixed exponentially growing populations). The authors consider 3 phases of human development: early development, growth and maintenance, and mature phase. In particular, time-dependent mutation rates in Figure 2d is an intriguing and strong result, and the process underlying Figures 3 and 4 are generally wellexplained and convincing.

Thank you for your positive comments.

Notes & suggestions:1. The explanation of Figure 2 in Lines 101 - 111 should be expanded for clarity. First, is Figure 2a derived from stochastic simulation (line 101 suggests) or some theoretical analysis? Second, the gradual transition from f-2 to f-1 is appreciated, but the shape of the intermediates is not addressed in detail. The power laws are straight lines, and the simulations provide curved lines -- please expand in what range (low or high frequency variants) the power law approximations apply.

Figure 2a was obtained from a numerical solution of equation 1, which describes the time dynamics of the expected VAF distribution. This is indeed unclear from the text, and we thank the reviewer for pointing out this discrepancy.

We thank the reviewer for this suggestion and have now adjusted this in the text (102-110):

“Numerical solutions of Eq.(1) show that the expected VAF distribution exhibits a gradual transition from the f-2 (growing population) to the f-1 (constant population) power law (Fig.2). These transitional states themselves do not adhere to some intermediate power-law (e.g. f^-α^ for 1<α<2), but instead present a sigmoidal shape, with the low frequency portion following f-1 and the high frequencies f-2 . Over time the shape changes as a wavelike front traveling from low to high frequency, with the constant-size equilibrium establishing earliest at the lowest frequencies and moving to higher frequency over time. Interestingly, the convergence towards equilibrium slows down over time -- for evenly-spaced observation times the solutions lie increasingly closer together -- further decreasing the speed at which the high frequency portion of the spectrum approaches equilibrium.”

We also changed the caption of Figure 2 to make this clearer as

“(a) Expected VAF distributions from evolving Eq1 to different time points for a population with an initial exponential growth phase and subsequent constant population phase (mature size N=103). Once the population reaches the maximum carrying capacity, the distribution moves from a 1/f2 growing population shape (purple) to a 1/f constant population shape (green). Note that the shift slows considerably at older age.”

In addition, we have also added annotations to Figure 2a and 2b to further clarify which line(green or purple) is f-1 and f-2.

Additionally, I do not understand the claim in line 108, that the transition is fast for low frequency variants, as the low frequency (on the left of the graph) lines are all close together, whereas the high frequency lines are far apart.

The lines are closer together in the low frequency portion (left of the plot) because they are already very close to the constant-size equilibrium (f-1/green line) and these frequencies approached equilibrium very fast. On the contrary, in the high frequency portion (right side of plot) they are still very far from equilibrium and approached equilibrium much slower.

It would be helpful to reiterate in this paragraph that these power laws are derived based on exponentially growing populations and are expected to break down under homeostatic conditions.

We have adjusted the relevant paragraph in the text to make the validity of the power laws clearer (90-94):

“For a well-mixed exponentially growing population without cell death the VAF spectrum v(f) is given by 2μ/(f+f2) (a f−2 power law) and is independent of time. In contrast, for a population of constant size – i.e. where birth and death rates are equal – the spectrum obeys v(f)∝2μ/f (a f−1 power law; see also SI), though this solution is only valid at sufficiently long times.”

1. The sample vs population (blue vs orange) in Figure 3 is under-explained. How is it that the mutational burden and inferred mutation rate in A and B roughly match, but the VAF distributions in C are so different? How was the sampled set chosen? Perhaps this is an unimportant distinction based on the particular sample set, but the divergence of the two in C may serve as a distraction, here.

This is an important question, and the answer was perhaps underemphasized in the caption. The sampling was performed as a uniform random sampling with replacement, and the same sample set was used for both the mutational burden and the VAF distribution. The reason for this stark contrast is that while the expectation of the burden distribution is not affected by sampling (i.e. sampling only affects the resolution/amount of stochasticity), the expectation of the VAF distribution changes due to sampling. While this was discussed in the section "Sparse sampling, single cell derived VAF spectra and evolutionary inferences", we have added note of this (indeed surprising) effect in the caption as well:

“(b) Distribution of estimated mutation rates from 10'000 individual simulations, obtained from burden distributions of the complete populations (blue) as well as sampled sets of cells (orange). Because the expected mutational burden distribution is unaltered by sampling, the expected estimate of the mutation rate from (5) remains unchanged: E(μ~pop)=E(μ~sample). However, sampling increases the noise on the observed burden distribution, which results in a higher errormargin of the estimate: σ(μ~pop)<σ(μ~sample).”

“(c) VAF spectra measured in the complete population (blue) and a sampled set of cells (orange). In contrast with the mutational burden distribution, strong sampling changes the shape of the expected distribution. A single simulation result is shown (diamonds) alongside the theoretically predicted expected values for both the total and sampled populations (Eqs. (1) and (6))(dashed line) and the average across 100 simulations (solid line).”

1. The comparison of results herein to claims by Mitchell (ref. 12) are quite important results within the paper. I appreciate the note in the final paragraph of the discussion, and I suggest adding a sentence referencing the result noted in line 248-249 to the abstract, as well.

We agree with the reviewer. We have extended the abstract now to reference the result in more detail:

“However, the single cell mutational burden distribution is over-dispersed compared to a model of Poisson distributed random mutations suggesting. A time-associated model of mutation accumulation with a constant rate alone cannot generate such a pattern. At least one additional source of stochasticity would be needed. Possible candidates for these processes may be occasional bursts of stem cell divisions, potentially in response to injury, or non-constant mutation rates either through environmental exposures or cell intrinsic variation.”

**Reviewer #2 (Public Review):**
Summary: The authors provide a nice summary on the possibility to study genetic heterogeneity and how to measure the dynamics of stem cells. By combining single cell and bulk sequencing analyses, they aim to use a stochastic process and inform on different aspects of genetic heterogeneity.Strengths: Well designed study and strong methods

Thank you for your positive comments.

Weaknesses: MinorFurther clarification to Figure 3 legend would be good to explain the 'no association' of number of samples and mutational burden estimate as per line 180-182 p.8.

We have added a note to the caption of Figure 3b to explain more clearly how sampling affects the burden distribution and the mutation rate inferred from it (see also previous response to Reviewer 1):

“Because the expected mutational burden distribution is unaltered by sampling, the expected estimate of the mutation rate from (5) remains unchanged: E(μ~pop)=E(μ~sample). However, sampling increases the noise on the observed burden distribution, which results in a higher errormargin of the estimate: σ(μ~pop)<σ(μ~sample).”

**Reviewer #1 (Recommendations For The Authors):**
Minor/editorial suggestions:1. Equation 1, please define \partial_t and \partial_K, for clarity.

These have now been defined in the text (between line 85-86): “where κ=fN(t) denotes the number of cells sharing a variant (the variant frequency f times the total population size N), δ(x) is the Dirac impulse function, ∂t and ∂κ are the partial derivatives with respect to time and variant size.”

1. Figure 2: It would be helpful to label the green and purple lines with the corresponding 1/f and 1/f^2 rule, in addition to the growing/fixed label, for clarity.

We agree and have now added the corresponding labels to each line.

**Reviewer #2 (Recommendations For The Authors):**
Minor suggestions are given below:It would be nice for the authors to comment on whether the results could be extended/modified to account for possible fitness advantage of mutations which would be clinically relevant, for instance in the case of CHIP mutations and difference in time to myeloid malignancies transformation between CHIP/No CHIP individuals.

This is an important point. We agree with the reviewer that CHIP mutations play an important role in shaping mutational diversity especially in older individuals. Evidence is now emerging that CHIP mutations are almost universally present in individuals 60+. Interestingly, in individuals younger than 60, a neutral model (as presented here), does capture the observed effective dynamics well. For the purpose of the analysis underlying this manuscript, a neutral model seems reasonable.

The techniques we use here can be adjusted to include selection. How the results extend or modify will critically depend on the actual model of selection (rare or frequent CHIP mutations, strong vs weak selection etc.) that is realized in human hematopoiesis. Here we would say, the underlying biology currently is mostly unknown and is subject to (by others and in part by us) ongoing investigations, which extend beyond the scope of this manuscript.

We now make note of this point in the manuscript and added a small paragraph in page 11 to the discussion:

“Another open question is the role of selection and how it shapes intra-tissue genetic heterogeneity. Evidence is emerging that positively selected variants in blood are almost universally present in individuals above 60, while the effective observable dynamics in younger individuals is well described by neutral dynamics. How results presented here generalize or modify will critically depend on the model of selection realized in human hematopoiesis, e.g. a models of rare or frequent driver events. Details of the underlying biology are currently unknown.”

It would be nice to see if any significant differences in parameter estimates occur between loci with/without linkage disequilibrium, for instance HLA region. Could the number of single-cell samples be 'more' relevant when studying the VAF distribution in HLA region?

This is a good suggestion. We might be wrong or missing an important point, but somatic evolution as we use it in our modeling here is solely driven by asexual reproduction of cells. As such the entire genome of the cell is in linkage disequilibrium, independent of the precise genomic region (somatic evolution is in first approximation blind to germline mutations, as they are present in every single cell of the organism and therefore do not carry any information on the somatic evolutionary dynamics).

We thank all editors and reviewers again for your constructive comments.